# From experimental clues to theoretical modeling: Evolution associated with the membrane-takeover at an early stage of life

**Wentao Ma**[1]*, Chunwu Yu[2]

**1** Hubei Key Laboratory of Cell Homeostasis, College of Life Sciences, Wuhan University, Wuhan, China,
**2** College of Computer Sciences, Wuhan University, Wuhan, China

* mwt@whu.edu.cn

## Abstract

Modern cell membranes are primarily composed of phospholipids, while primitive cell membranes in the beginning of life are believed to have formed from simpler lipids (such as fatty acids) synthesized in the prebiotic environment. An attractive experimental study suggested that the corresponding "membrane-takeover" (as an evolutionary process) is likely to have occurred very early (e.g., in the RNA world) due to some simple physical effects, and might have subsequently triggered some other evolutionary processes. Here, via computer modeling on a system of RNA-based protocells, we convinced the plausibility of such a scenario and elaborated on relevant mechanisms. It is shown that in protocells with a fatty-acid membrane, because of the benefit of phospholipid content (i.e., stabilizing the membrane), a ribozyme favoring the synthesis of phospholipids may emerge; subsequently, due to the reduced membrane permeability on account of the phospholipid content, two other functional RNA species could arise: a ribozyme exploiting more fundamental materials (thus more permeable) for nucleotide synthesis and a species favoring across-membrane transportation. This case exemplifies a combination of experimental and theoretical efforts regarding early evolution, which may shed light on that notoriously complicated problem: the origin of life.

## Author summary

Based on both logical reasoning and relevant evidence, protocells in the origin of life should have had a membrane composed of simple lipids like fatty acids. An interesting lab study suggested that the primitive membrane might have evolved towards a phospholipid membrane like that in modern cells quite early, due to phospholipids' role of stabilizing the membrane. That is, the protocells with a function of facilitating phospholipid synthesis may have been favored in the competition and thus overwhelmed other protocells. Then, for such protocells,

**Data availability statement:** Source codes of the simulation program can be obtained from: https://github.com/mwt2001gh/membrane-takeover/blob/main/Fig-gr-final-1.cpp (corresponding to the case shown in Fig 2a) and https://github.com/mwt2001gh/membrane-takeover/blob/main/Fig-npr-final-1.cpp (corresponding to the case shown in Fig 6a). All other relevant data are in the manuscript and its supporting information files.

**Funding:** This study was supported by the National Natural Science Foundation of China (No. 31571367) (http://www.nsfc.gov.cn) and Natural Science Foundation of Hubei Province (CN) (No. 2019CFB685) (http://kjt.hubei.gov.cn) to WM. The funders had no role in study design, data collection and analysis, decision to publish, or preparation of the manuscript.

**Competing interests:** The authors have declared that no competing interests exist.

which have a less permeable membrane due to higher phospholipid content, selective pressure may have further driven the emergence of the functions of utilizing more permeable raw materials and improving the across-membrane transportation. Indeed, concerning early evolution occurring during the origin of life, experiments may offer valuable clues, but usually have difficulty in providing insights into the corresponding process. On the other hand, computer modeling (simulation) has become a powerful tool for us to investigate relevant scenarios. Here, by modeling we demonstrated the suggested evolutionary process and investigated relevant mechanisms. The combination of experimental efforts and theoretical modeling efforts like this is expected to be an effective way for us to explore the complicated early evolution.

## Introduction

It is thought that the primitive cells (protocells) should have had a membrane composed of simple, single-chain lipids, such as fatty acids and their derivatives (generally referred to as "fatty acids" below for concise), which were present in the prebiotic environment [1–3]. Indeed, though phospholipids may also have been synthesized prebiotically [4,5], they, as more complex molecules, are likely to have been much less abundant – especially considering that they should have been made from those single-chain lipids. In the logic of "the simpler, the more likely to emerge *de novo*", it is more plausible that the first membranes were assembled from the single-chain lipids, and phospholipid membranes came later, perhaps due to Darwinian evolution. Another major reason in favor of "fatty acids first" for primordial membranes is that phospholipid-based membranes are much less permeable, thus would seriously hinder protocells from obtaining crucial materials available in environments for growth and reproduction [1–3].

Traditionally, as a further argument for the scene of "fatty acids first", it is emphasized that these single-chain lipids are more dynamic – with small, less polar head groups, their exchange between leaflets (the so-called flip-flop) is much faster, thus favoring the spontaneous growth and division of protocells [1–3]. But evidence has now accumulated to show that phospholipid-based membranes are also sufficiently dynamic for growth and division [4,6–8]. Therefore, this argument against phospholipids being the first membrane lipids seems no longer that strong. However, notably, if the scene of "fatty acids first" is true, these findings on phospholipid-based membranes actually imply that the membrane-takeover from fatty-acid membranes to phospholipid membranes could have occurred quite early – well before the advent of complex forms like modern cells, which have specific functions regarding cellular growth and division.

Interestingly, via experimental studies, Budin and Szostak found evidence in support of such an early membrane-takeover [9]. It was shown that fatty acid vesicles with a portion of phospholipid components would grow at the expense of those pure fatty acid vesicles (see also associated earlier work [10,11]). The main reason is that

the involvement of phospholipids would reduce the efflux of fatty acids from the membrane and, through the exchange equilibrium of these molecules between vesicles via the environment, eventually result in a net inflow of fatty acids. This means, as the authors stated, "the ability to synthesize phospholipids from single-chain substrates would have therefore been highly advantageous for early cells competing for a limited supply of lipids" [9]. Undoubtedly, with the emergence of this function in protocells, the phospholipid content of the membrane would have increased.

Furthermore, in the same work, it was demonstrated that the permeability of the membrane declines in proportion to the rise in phospholipid content [9]. The reduction of permeability was ascribed to the decreased fluidity (i.e., increased order) in a membrane contain more phospholipid molecules (in the form of double-chain lipids). This "would have led to a cascade of new selective pressures for the evolution of metabolic and transport machinery to overcome the reduced membrane permeability" [9]. In other words, as they clarified, "cells could have evolved the ability to synthesize their own building blocks from simpler, more permeable substrates", and "membrane transporters, a hallmark of modern cells, would have emerged as a means for overcoming low membrane permeability".

Obviously, by exploring relevant physical effects, the experimental work was trying to formulate speculations on some tendencies of Darwinian evolution in the early stage of life. However, at least up to now, such evolutionary tendencies are *per se* difficult to follow up in laboratory. On the other hand, with the rapid development of theoretical studies in the field of the origin of life (e.g., see refs [12]–[16]), it is now well feasible to model such early evolution, i.e., to study relevant evolutionary dynamics by computer simulation. So here we ask: can we demonstrate *in silico* the plausibility of the evolutionary events suggested by the experimental work? Additionally, via the modeling, it is expected that we would get a more comprehensive understanding on detailed mechanisms involved in the evolution.

The RNA world hypothesis is now widely accepted in the field of the origin of life [17–19], due to its logical reasonability as well as accumulating evidence supporting it. In fact, the most meaningful point of this idea is that it tries to explain the onset of Darwinian evolution [16,20,21] – and the subsequent process in life's history is just a matter of Darwinian evolution. In the scenario, RNA played both the roles of genetic material and functional molecules, as the two fundamental requirements for the "running" of Darwinian evolution (thus evading the "Chicken and Egg" dilemma – which came first, DNA or proteins?). Though there are still ongoing debates on it, the scenario undoubtedly offers a relatively simple platform for us to model early evolutionary events like the ones we are concerned about here (actually, in the original experimental work it was also implied that the suggested functions associated with the membrane-takeover may have evolved in the RNA world [9]). Therefore, here we aim to model the emergence of a ribozyme favoring the synthesis of phospholipids in RNA-based protocells – due to the phospholipids' benefit for stabilizing the membrane, and the subsequent arising of a ribozyme favoring the exploitation of simpler (thus more permeable) substrates, or that of an RNA functional species favoring the membrane transport – owing to the decreased membrane permeability resulting from the increased phospholipid content.

## Results

### About the model

We conducted the computer simulation using a Monte Carlo model similar to those used in our previous work concerning the RNA-based protocells [22–24]. It is described below in general terms (see Methods for details). The system is a two-dimensional $N \times N$ square grid (with toroidal topology to avoid edge effects). Molecules are distributed within the grid rooms, including nucleotides, RNA, fatty acids, phosphatidic acids (here as a representative of phospholipids) and glycerophosphates (the head-group-maker of phosphatidic acids), as well as some relevant precursors: nucleotide precursors, nucleotide-precursor's precursors, and glycerophosphate precursors. Amphiphiles (fatty acids and phospholipids) may assemble at the boundary of a grid room and form a membrane, then the grid room is occupied by a protocell. In each time step, certain events may occur to molecules and protocells with defined probabilities (Table 1).

**Table 1. Parameters used in the computer simulation.**

| Probabilities | Descriptions | Default Values |
|---|---|---|
| $P_{AT}$ | An RNA template attracting a substrate (by base-pairing) | 0.9 |
| $P_{BB}$ | A phosphodiester bond breaking within an RNA chain | $1 \times 10^{-5}$ |
| $P_{CB}$ | A protocell breaking | $1 \times 10^{-4}$ |
| $P_{CD}$ | A protocell dividing | 0.1 |
| $P_{CF}$ | Two adjacent protocells fusing with each other | 0.001 |
| $P_{FJM}$ | A fatty acid joining the membrane | 0.9 |
| $P_{FLM}$ | A fatty acid leaving the membrane | 0.002 |
| $P_{FP}$ | The false base-pairing when a template attracts a substrate | 0.001 |
| $P_{GD}$ | A glycerophosphate decaying into its precursor | 0.1 |
| $P_{GF}$ | A glycerophosphate forming from its precursor (non-enzymatic) | 0.002 |
| $P_{GFR}$ | A glycerophosphate forming from its precursor catalyzed by GR | 0.9 |
| $P_{GPP}$ | A glycerophosphate precursor permeating through the membrane | 0.9 |
| $P_{MC}$ | A protocell moving | 0.1 |
| $P_{MF}$ | A membrane forming | 0.1 |
| $P_{MV}$ | A nucleotide/fatty acid/phospholipid (or relevant precursors) moving | 0.9 |
| $P_{ND}$ | A nucleotide decaying into its precursor | 0.02 |
| $P_{NDE}$ | A nucleotide residue decaying at RNA's chain end | 0.001 |
| $P_{NF}$ | A nucleotide forming from its precursor (non-enzymatic) | 0.005 |
| $P_{NFR}$ | A nucleotide forming from its precursor catalyzed by NR | 0.2 |
| $P_{NP}$ | A nucleotide permeating through the membrane | $5 \times 10^{-5}$ |
| $P_{NPD}$ | A nucleotide precursor decaying into its precursor | 0.005 |
| $P_{NPF}$ | A nucleotide precursor forming from its precursor (non-enzymatic) | 0.002 |
| $P_{NPFR}$ | A nucleotide precursor forming from its precursor catalyzed by NPR | 0.3 |
| $P_{NPP}$ | A nucleotide precursor permeating through the membrane | 0.05 |
| $P_{NPPP}$ | A nucleotide-precursor's precursor permeating through the membrane | 0.5 |
| $P_{PD}$ | A phospholipid decaying (into fatty acid and glycerophosphate) | 0.1 |
| $P_{PDM}$ | A phospholipid decaying within the membrane | 0.01 |
| $P_{PF}$ | A phospholipid forming (on the membrane) | 0.02 |
| $P_{PJM}$ | A phospholipid joining the membrane | 0.9 |
| $P_{PLM}$ | A phospholipid leaving the membrane | $1 \times 10^{-4}$ |
| $P_{RL}$ | The random ligation of nucleotides and RNA | $1 \times 10^{-6}$ |
| $P_{SP}$ | The separation of a base pair | 0.5 |
| $P_{TL}$ | The template-directed ligation of RNA | 0..02 |
| **Others** | **Descriptions** | **Default Values** |
| $N$ | The system is defined as an $N \times N$ grid | 30 |
| $T_{NPPB}$ | Total nucleotide-precursor's precursors introduced in the beginning | 80000 |
| $T_{FB}$ | Total fatty acids introduced in the beginning | 50000 |
| $T_{GPB}$ | Total glycerophosphate precursors introduced in the beginning | 50000 |
| $F_{DE}$ | Factor of the Donnan's equilibrium effect | 1 |
| $F_{DO}$ | Factor of molecular degradation outside protocells | 10 |
| $F_{PL}$ | Factor of phospholipids' influence on amphiphiles leaving the membrane | 5 |
| $F_{PP}$ | Factor of phospholipids on permeability (for nucleotides or their precursors) | 20 |
| $F_{PPW}$ | $F_{PP}$ for nucleotide-precursor's precursors or glycerophosphate precursors | 3 |
| $F_{TR}$ | Factor for the RNA species functioning in membrane transport (TR) | 100 |
| $L_{AM}$ | The lower limit number of amphiphiles to form a membrane | 200 |
| $L_{CDR}$ | The length of the characteristic domain of a functional RNA species | 7 |

Note: The probabilities are listed with names in alphabetical order. The simulation cases shown in this paper adopt the default values, unless being stated explicitly to be different. The default characteristic sequence for GR is "UUGAGCG", for NR is "GCACGUA", for NPR is "UCACGAG", for TR is "CUGCUAG", and for the control is "GGCUACU". See Methods for details on the principle of setting parameter values.

In the beginning of a simulation, a certain quantity of nucleotide-precursor's precursors, fatty acids and glycerophosphate precursors are introduced into the system. During the simulation process, protocells or RNA species may be inoculated (see below for detailed descriptions in different cases). In the system, nucleotide-precursor's precursors may transform into nucleotide precursors, which in turn form nucleotides (randomly as A, G, C, or U). Nucleotides may assemble into RNA via random ligation. RNA may conduct template-directed replication. Glycerophosphate precursors may transform into glycerophosphates, which in turn react with fatty acids on the membrane and produce phosphatidic acids thereon. With its membrane absorbing amphiphiles, a protocell may grow and after reaching a certain size, may divide into two – the molecules within it and the amphiphiles on its membrane would assort randomly into its offspring (thus reproduction).

Significantly, the model has a "resolution" at the nucleotide level, thus inherently suitable for studying the early Darwinian evolution, which relies essentially on the sequence-function connection. In the model, an RNA molecule containing a characteristic sequence (domain) is assumed to have a special function (i.e., as a ribozyme). The total materials (for RNA and the membrane) in the system is constant, and the RNA-based protocells compete for these materials. In the competition, those protocells containing functional RNA species "beneficial" to protocell's reproduction may spread (become thriving) in the system – or say, those "useful" functional RNA species may spread among protocells. In practice, here the characteristic sequence of a functional RNA species is arbitrarily presumed on account of our ignorance of relevant cases, but this does not matter – what our modeling aims to explore is merely: if a characteristic sequence bears a special function, can the sequence spread? Or more abstractly, could a specific "sequence-function connection" result in a case of Darwinian evolution?

### The spread of the ribozyme favoring phospholipid-synthesis in protocells

As suggested by the original experimental work [9], the ability to synthesize phospholipids would have been highly advantageous for protocells competing for a limited supply of lipids, since the efflux of fatty acids would decrease with the increase of phospholipid content in the membrane. As mentioned above regarding the model, we consider phosphatidic acids as a representative of phospholipids here. In reality, phosphatidic acid is the simplest phospholipid and was indeed likely to have been directly involved in the membrane-takeover. In the potential relevant synthetic route, phosphorylation of glycerol appears to have been inefficient, while the acylation of glycerophosphates by fatty acids could have been productive [5,25,26]. In other words, the bottleneck was the formation of glycerophosphates – thus, here we assume a glycerophosphate-synthetase ribozyme (GR) as a representative of the supposed "ribozyme favoring phospholipid-synthesis", and the synthesized glycerophosphate molecules would reach the protocell membrane and react with fatty acids therein in a non-enzymatic way. Indeed, a recent study demonstrated that the acylation leading to phospholipids may well have occurred free of enzymes [4]. Fig 1 shows a scheme depicting how a protocell containing GR could have grown at the expense of the one without GR. With the growth of membrane, GR within the protocell may replicate; with the enlargement of the protocell, it may divide into offspring protocells due to physical instability [3,27] – thereby achieving "reproduction". The protocell without GR would shrink and may eventually break or fuse with other protocells.

First of all, we want to explore whether protocells containing GR could become thriving by virtue of its function in favoring the synthesis of phospholipids. In the simulation, an "empty" fatty-acid protocell is inoculated at step $1 \times 10^3$. By absorbing fatty acids in the system, the empty protocells grow and divide – eventually spreading in the system. Then, at step $1 \times 10^4$, ten empty protocells are selected (arbitrarily, the same below), each inoculated with one GR molecule, while ten other empty protocells are each inoculated with one control (RNA species without function) molecule. It was found that the protocells containing GR could spread, whereas the ones containing the control could not (Fig 2a, the upper panel); in other words, the GR could spread among protocells, whereas the control could not (Fig 2a, the lower panel).

In order to study the underlying mechanism, we investigated the influences of several key parameters. Firstly, to confirm that the spread of GR protocells (or say, the spread of GR) is owing to the function of GR, we explored the influence of $P_{GFR}$ (the probability of glycerol-phosphate formation catalyzed by the ribozyme; see Table 1 for descriptions of

PLOS Computational Biology

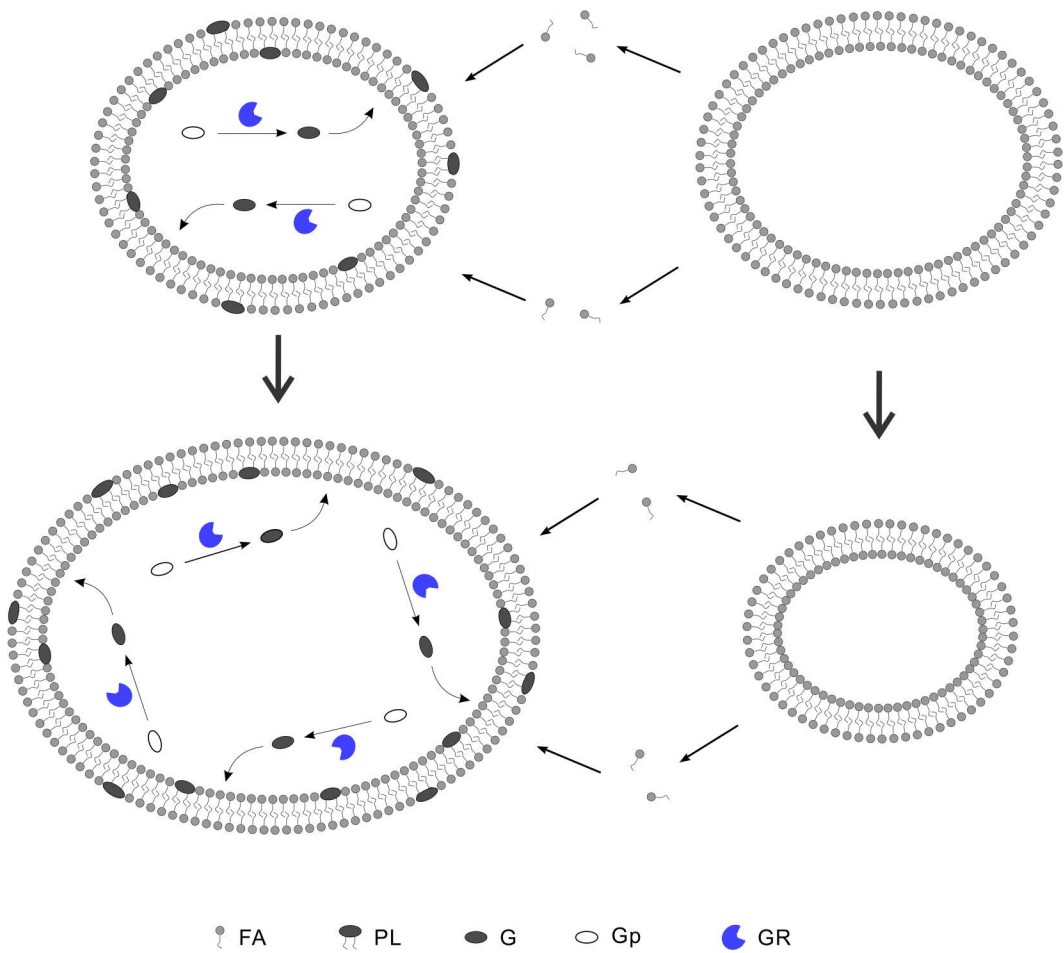

FA    PL    G    Gp    GR

**Fig 1. Protocells would benefit from a ribozyme favoring the synthesis of phospholipids in the competition.** Legends: FA—fatty acid; PL—phospholipid (i.e., phosphatidic acid here); G—glycerophosphate; Gp—glycerophosphate precursor (e.g., glycerol); GR—glycerophosphate-synthetase ribozyme (here representing the ribozyme favoring the synthesis of phospholipids). The glycerophosphates produced through the catalysis of GR may reach the membrane and non-enzymatically react with fatty acids therein to form phosphatidic acids (the phospholipid molecules synthesized on the inner layer of the membrane may flip to the outer layer). The formation of phospholipids on the membrane would prevent fatty acids from leaving the membrane to a certain extent, which results in a net inflow of fatty acids in the lipid competition. With the growth of the membrane, the number of GR within the protocell may increase as a result of RNA replication.

parameters). Indeed, with the stepwise turning-down of the ribozyme function, the spread of the GR protocells is weakened and finally completely suppressed (Fig 3-$P_{GFR}$).

Next, we were interested in whether the advantage of GR protocells could be attributed to the decrease of fatty acid desorption from their membranes, as suggested by the original experimental work [9]. In accordance with the experimental work, the probability of a fatty acid molecule leaving the membrane is here assumed to be negatively correlated with the content of phospholipids in the membrane, i.e., in proportion to $1/(1+F_{PL} \times \text{RPM})$, where RPM is the ratio of phospholipids in the membrane and $F_{PL}$ is a factor representing the degree of this influence (see Methods for details). Somewhat surprisingly, the decrease of $F_{PL}$ does not significantly affect the spread of GR protocells (Fig 3-$F_{PL}$, cyan symbols) – even when $F_{PL}$ is set to 0 (after step $2.5 \times 10^6$), which indicates that phospholipids in the membrane no longer have an impact on the desorption of fatty acids, the spread of GR protocells is only marginally inhibited. That is to say, there should be other mechanisms that favor the GR protocells.

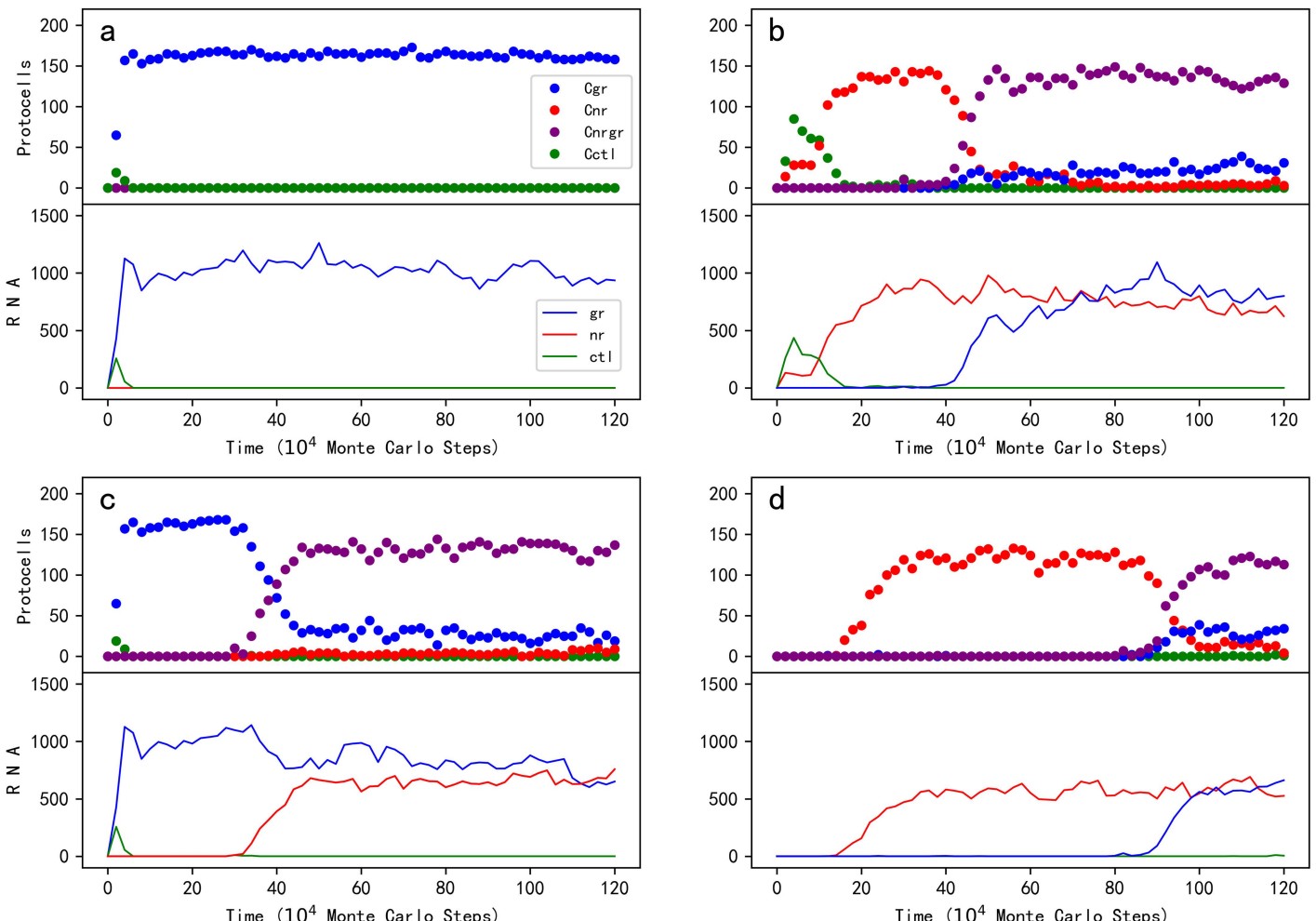

**Fig 2. The spread of the glycerophosphate-synthetase ribozyme (GR) and its co-spread with the nucleotide-synthetase ribozyme (NR) in RNA-based protocells.** For each subfigure, the upper panel shows the trend of protocells containing specific RNA species while the lower panel demonstrates the trend of the total molecule number of the relevant RNA species in the system. Legends: Cgr — protocells containing GR; Cnr — protocells containing NR; Cnrgr — protocells containing NR and GR; Cctl — protocells containing the control RNA species; gr — GR; nr — NR; ctl — the control RNA species (the legends apply to all the subfigures). For all the cases, an "empty" fatty-acid protocell is inoculated at step $1 \times 10^3$. (a) The *de novo* spread of GR among protocells. Wherein, at step $1 \times 10^4$, ten empty protocells are selected (arbitrarily, the same below), each inoculated with one GR molecule, and another ten empty protocells are selected, each inoculated with one control molecule. (b) The spread of GR in protocells containing NR. Wherein, at step $1 \times 10^4$, ten empty protocells are selected, each inoculated with one NR, and another ten empty protocells are selected, each inoculated with one control; at step $3 \times 10^5$, ten NR protocells are selected, each inoculated with one GR, and another ten NR protocells are selected, each inoculated with one control. (c) The spread of NR in protocells containing GR, Wherein, at step $1 \times 10^4$, ten empty protocells are selected, each inoculated with one GR, and another ten empty protocells are selected, each inoculated with one control; at step $3 \times 10^5$, ten GR protocells are selected, each inoculated with one NR, and another ten GR protocells are selected, each inoculated with one control. (d) An evolutionary case without inoculation of the RNA species – first, NR occurs naturally in empty protocells, and then GR occurs naturally in NR protocells. $P_{RL} = 5 \times 10^{-6}$. The characteristic sequence of GR is "CCAUGUA" – only two nucleotides different from that of NR (default sequence: "GCACGUA", see the footnotes of Table 1); the control species adopts a characteristic sequence of "UCAGGUA", two nucleotides different from either of the two ribozymes.

In fact, in that original paper [9], a potential additional reason was proposed: with a fraction of "insoluble" phospholipids, actually, "only the fraction of the vesicle surface area composed of fatty acids can contribute to monomer efflux, whereas the entire surface area permits fatty acid influx, leading to a net influx (growth)". In other words, the formation of phospholipid molecules from fatty acids on the membrane could "fasten" this portion of fatty acids. In our model, the default value

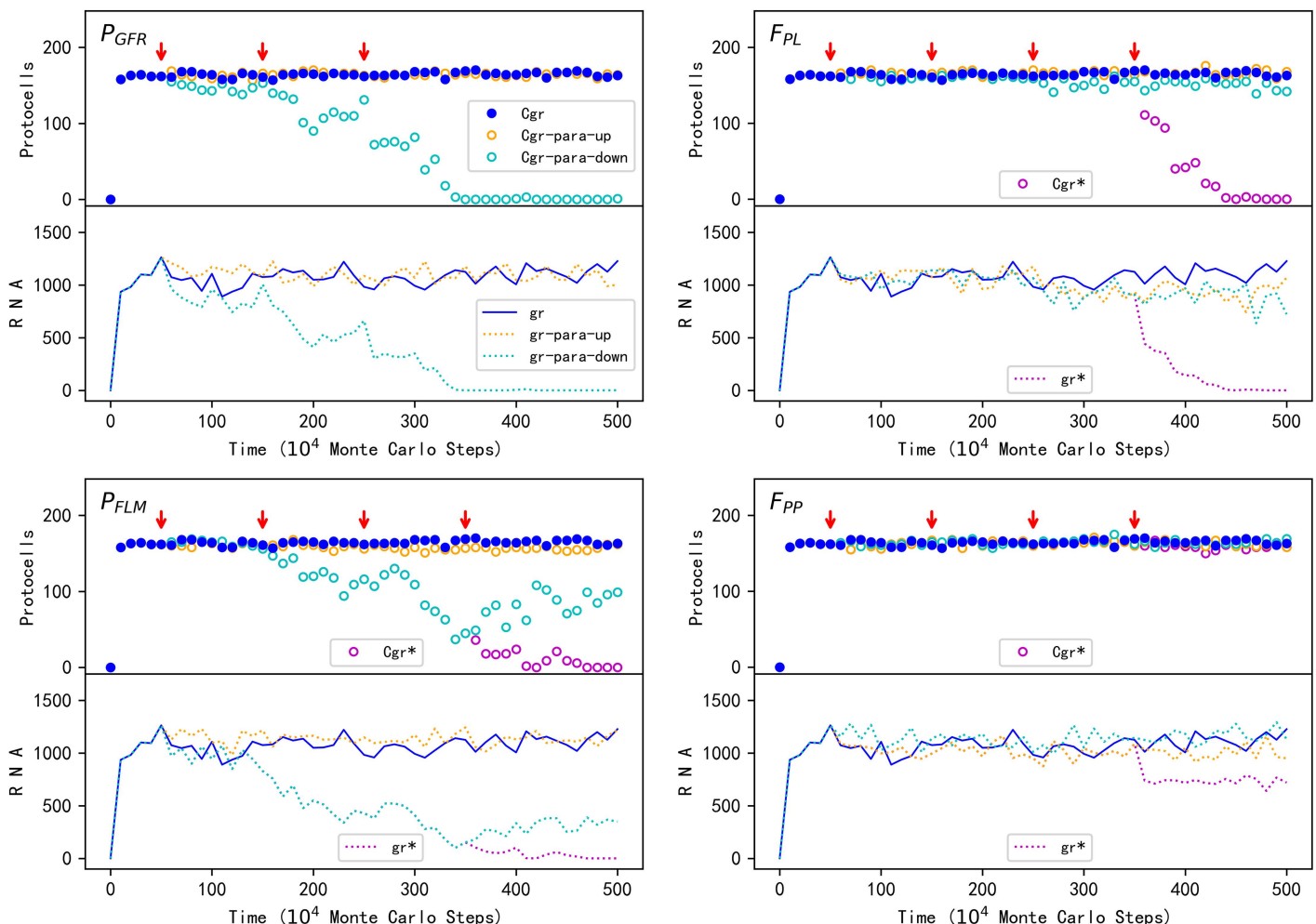

**Fig 3. The influence of several key parameters on the spread of the protocells containing GR.** For each subfigure, the upper panel shows the trend of GR protocells while the lower panel demonstrates the trend of the total number of GR molecules in the system. Legends: Cgr — GR protocells with all parameters adopting default values; Cgr-para-up — GR protocells with the relevant parameter increasing; Cgr-para-down — GR protocells with the relevant parameter decreasing; gr — GR with all parameters adopting default values; gr-para-up — GR with the relevant parameter increasing; gr-para-down — GR with the relevant parameter decreasing (the legends apply to all the subfigures). The red arrows indicate the critical steps where the parameter adjustments are conducted. For ($P_{GFR}$), the default value 0.9 is turned up to 0.95, 0.98 and 0.99 at these points of change, respectively, or turned down to 0.1, 0.05 and 0.02 at these points. For ($F_{PL}$), the default value 5 is turned up to 10, 20 and 50 at the first three points of change, respectively, or turned down to 2, 1 and 0 at these points; additionally, at the fourth change point of the turning-down case, $P_{FLM}$ is changed from its default value 0.002 to $1 \times 10^{-4}$ (the legends Cgr* and gr* refer to this change). For ($P_{FLM}$), the default value 0.002 is turned up to 0.005, 0.01 and 0.02 at the first three points of change, respectively, or turned down to $5 \times 10^{-4}$, $2 \times 10^{-4}$ and $1 \times 10^{-4}$ at these points; additionally, at the fourth change point of the turning-down case, $F_{PL}$ is changed from its default value 5 to 0 (the legends Cgr* and gr* refer to this change). For ($F_{PP}$), the default value 20 is turned down to 10, 5 and 2 at the first three change points, respectively, or turned up to 200, 2000 and $2 \times 10^4$ at these points; additionally, at the fourth change point of the turning-up case, $F_{PPW}$ is changed from its default value 3 to 3000 (the legends Cgr* and gr* refer to this change).

of the probability of a phospholipid molecule leaving the membrane ($P_{PLM} = 1 \times 10^{-4}$) is much lower than that for a fatty acid molecule ($P_{FLM} = 0.002$). Therefore, in addition to set $F_{PL}$ to 0, we tried to assume the same value for these two probabilities (thus the "fastening effect" no longer exists) – and "witnessed" the collapse of GR protocells' spread! (Fig 3-$F_{PL}$, purple symbols, where after step $3.5 \times 10^6$ $P_{FLM}$ is set to $1 \times 10^{-4}$; see also Fig A in S1 Text, purple symbols, where after step $3.5 \times 10^6$ $P_{PLM}$ is increased to 0.002). Subsequently, we explored the "fasten effect" *per se* – indeed, the decreasing of $P_{FLM}$

(thus more approaching to the value of $P_{PLM}$) comes against the spread of GR protocells (Fig 3-$P_{FLM}$, cyan symbols). In this case, even when $P_{FLM}$ is set to a value identical with that of $P_{PLM}$ (after step $2.5 \times 10^6$), the GR protocells can still spread at a significant level, which is then completely suppressed when $F_{PL}$ is set to 0 (after step $3.5 \times 10^6$, purple symbols). That is to say, the two reasons mentioned above, the "fastening effect" and the "anti-desorption effect", do work together – both contribute to the net influx of fatty acids, and eventually result in the spread of GR protocells.

### The co-spread of the ribozyme favoring phospholipid-synthesis and another ribozyme in protocells

After observing that GR may spread among protocells *de novo*, we asked whether this phospholipid-synthesis favoring ribozyme may become thriving in protocells that already contain other ribozymes. An RNA species catalyzing the template-directed synthesis (and thus the RNA replication) has long been suggested to have been the first ribozyme emerging in the RNA world, usually referred to as an "RNA replicase" [18,28–31]. Another appealing candidate is a ribozyme capable of catalyzing the synthesis of nucleotides [32–34], namely nucleotide-synthetase ribozyme (NR) – it may also favor its own replication by supplying monomers (the replication could have based on non-enzymatic copying of RNA) [35,36]. In fact, both the two ribozymes, as supported by modeling work, might have spread early in the RNA world [24,35,37–40]. Here we choose NR-containing protocells as the target protocells to see whether GR could spread therein, mainly considering that in this study we will later introduce a related ribozyme, i.e., a nucleotide-precursor-synthetase ribozyme (NPR) – it is attractive to involve two ribozymes in the same "pathway".

In the simulation, after the initial inoculation at step $1 \times 10^3$, empty protocells spread; then, at step $1 \times 10^4$, ten of them are selected, each inoculated with one NR molecule (and control molecules are inoculated into another ten empty protocells). As expected, the NR protocells spread (while the protocells with the control cannot) (Fig 2b). Subsequently, at step $3 \times 10^5$, ten NR protocells are selected, each inoculated with one GR, and another ten NR protocells are each inoculated with one control. Eventually, protocells containing both NR and GR spread in the system – or say, GR co-spreads with NR among protocells (while the control cannot).

Noticeably, in his later essays, the leader of the original experimental work, Prof. Szostak explicitly suggested that a ribozyme favoring phospholipid-synthesis might have emerged first in the RNA world, and other beneficial ribozymes followed [36,41]. Therefore, based on the case shown in the *de novo* spread of GR (Fig 2a), we investigate whether NR could follow. After the spread of GR protocells, ten of them are selected, each inoculated with one NR (another ten GR protocells are each inoculated with one control). Eventually, protocells containing both NR and GR spread in the system (Fig 2c).

In the three cases mentioned above, to avoid the influence of random events such as RNA degradation, we investigated the plausibility of the spread of GR or the co-spread of GR and NR through selecting ten protocells and inoculating each with one molecule of relevant RNA species. The observed spread or co-spread, in fact, already reflects a full sense of "Darwinian evolutionary dynamics", meaning that in reality, once the RNA species appeared, they may have become thriving among protocells. As an example, in Fig 2d, we showed an instance of evolution without any inoculation of ribozyme. Firstly, NR emerges naturally in empty protocells, and then GR emerges naturally in NR protocells (see Fig 4 for snapshots on the spatial distribution during the evolution). In this case, to "facilitate" the natural appearance of a ribozyme, the probability of random ligation of RNA ($P_{RL}$) is augmented from its default value $1 \times 10^{-6}$ to $5 \times 10^{-6}$; and to "promote" the natural appearance of the other ribozyme in protocells contained the first ribozyme, their characteristic sequences are assumed as different from each other with only two residues (see the figure's legends for details).

### About the membrane takeover and the influence of decreased permeability

Then we were concerned about the change of membrane contents resulting from the spread of the ribozyme favoring phospholipid-synthesis. Fig 5a shows the membrane change coming with the spread of GR in empty protocells (the case is just the one shown in Fig 2a; but here in order to demonstrate the transition clearly, the horizontal axis adopts a

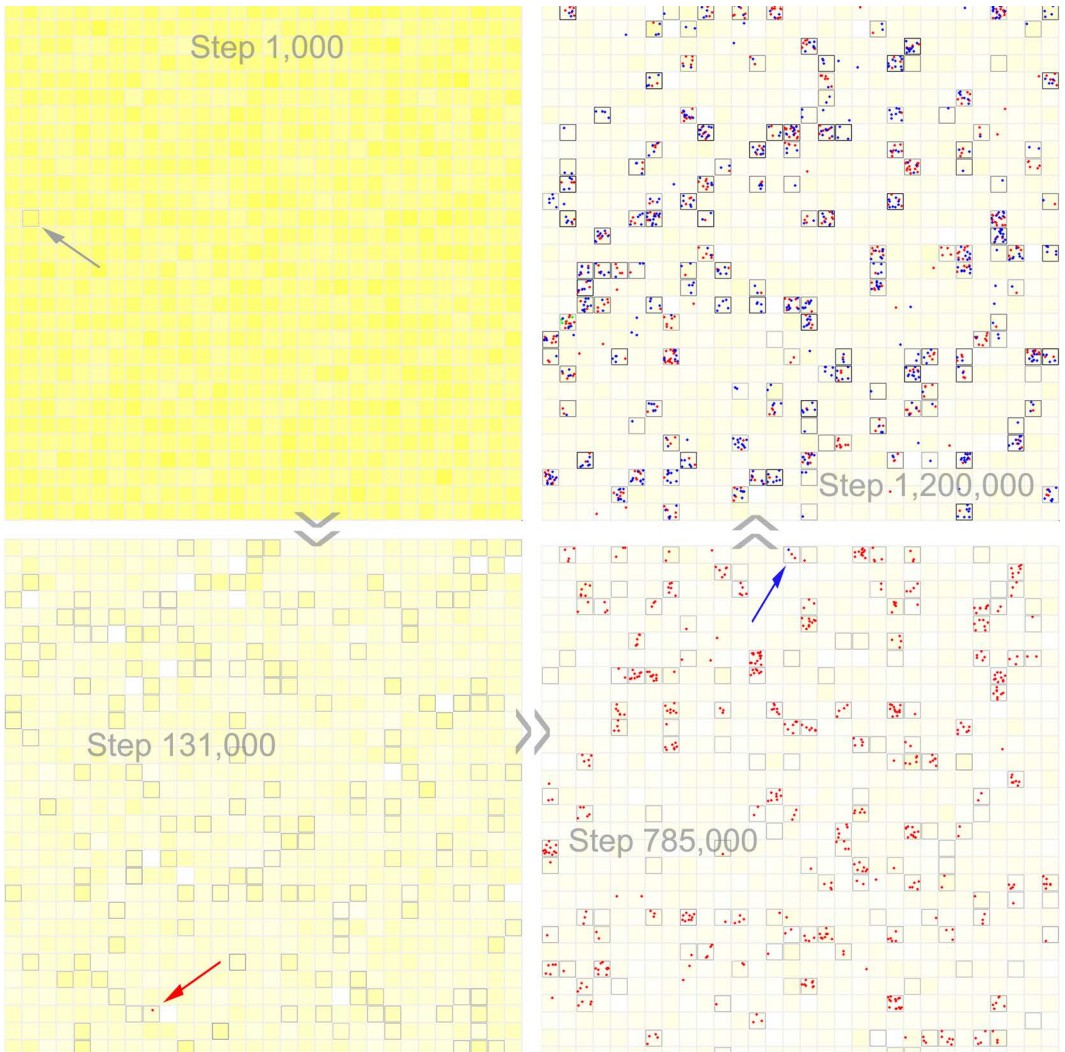

**Fig 4. The snapshots on spatial distribution of a case exemplifying the natural spread of NR and GR among protocells.** The evolutionary dynamics of the case is shown in Fig 2d. The color-depth of yellow in the background represents the concentration of the raw materials for forming nucleotides in the system (i.e., precursors of nucleotide precursors). The grey squares denote the membranes of protocells, and the corresponding color-depth is in proportion to the phospholipid content in the membrane. The red dots denote NR, and the blue dots denote GR. An empty protocell is inoculated at step 1000 (the grey arrow), and then empty protocells spread in the system (in reality, the first empty protocell might have formed due to the inducing of mineral particles [69] or the concentration effect during dry-wet circles [61,62]). The red arrow indicates the first NR emerging naturally in an empty protocell. The blue arrow indicates the first GR emerging naturally in an NR protocell.

smaller scale). Fig 5b shows the membrane change coming with the spread of GR in NR protocells (the case is just the one shown in Fig 2b). In both cases, we can see a rising of the ratio of phospholipids in the membrane (RPM, the lower panel) for the protocells with GR (i.e., Cgr in Fig 5a and Cnrgr in Fig 5b). Note that the RPM for those protocells without GR (i.e., C in Fig 5a and Cnr in Fig 5b) also increases a bit, which may be attributed to phospholipids' exchange between protocells.

No matter how, we can see that in both cases, RPM only rises to a limited level after the emergence of GR. We reasoned that this should be on account of the relatively inefficient non-enzymatic reaction of glycerophosphates with fatty acids on the

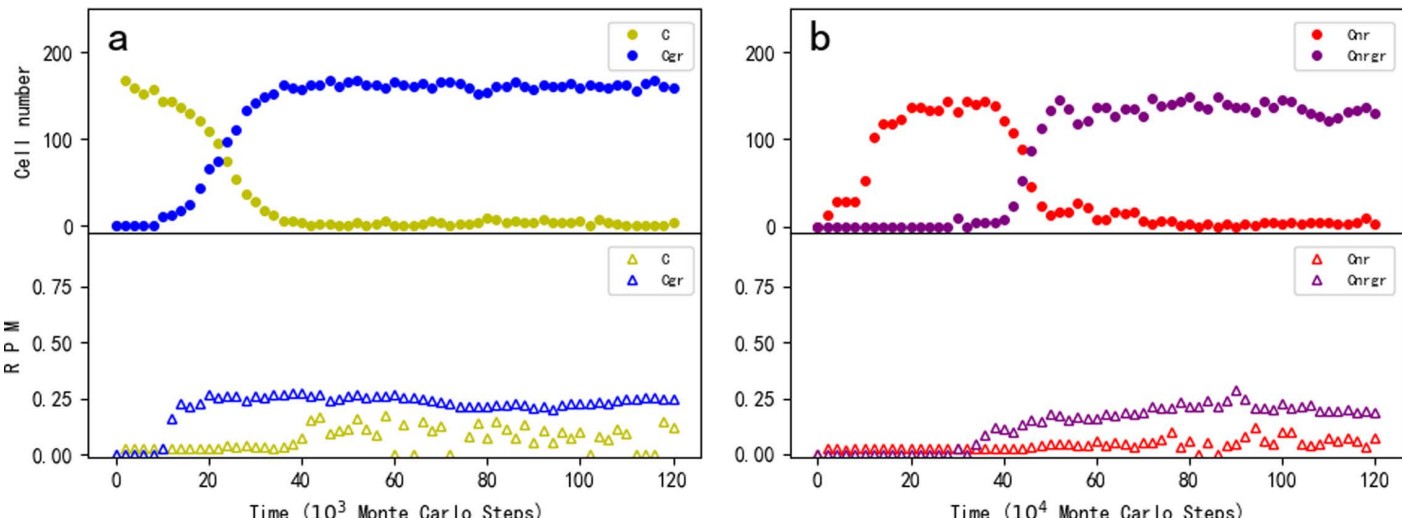

**Fig 5. The alteration of membrane contents with the rising of a ribozyme favoring phospholipid-synthesis (i.e., GR).** For each of the two sub-figures, the upper panel shows the trend of relevant protocells while the lower panel demonstrates the trend of RPM (the ratio of phospholipids in the membrane) for these protocells (averaged for each type of protocell). Legends: C — empty protocells; Cgr — protocells containing GR; Cnr — protocells containing NR; Cnrgr — protocells containing NR and GR. RPM for a protocell is calculated as $2 \times p_{num}/(2 \times p_{num} + f_{num})$, where $p_{num}$ and $f_{num}$ denote the number of phospholipids and that of fatty acids on the membrane respectively (note that a phospholipid molecule has two non-polar tails whereas a fatty acid has one). Here the vertical axis represents the average RPM of the corresponding protocells, and it is set to 0 *ad hoc* at the points where that kind of protocells does not exist. (a) The change of membrane contents during the spread of GR in empty protocells (the case is the same as the one shown in Fig 2a, but the horizontal axis adopts a smaller scale). (b) The change of membrane contents during the spread of GR in NR protocells (the case is the same as the one shown in Fig 2b).

membrane. That is, while glycerophosphates are plenty due to the function of GR, the subsequent reaction of phospholipids' formation becomes the bottle neck. Indeed, when we assume a high rate for the non-enzymatic formation of phospholipids (i.e., $P_{PF}$) in the midway, the ratio of phospholipids rises to a rather high level immediately (Fig B in S1 Text). This result implies that in reality, it should be the later emergence of a ribozyme or enzyme favoring this bottle-neck reaction that may have ultimately taken the membrane-takeover towards a more thorough degree. Remarkably, merely via inducing such a "limited membrane takeover" (i.e., with a low level of phospholipid content in the membrane), GR is capable of thriving.

Since the phospholipid content in the membrane would reduce the membrane permeability [9] and thus the availability of raw material for protocells, the GR's advantage might be weakened. It was then attractive to see how the GR protocells' spread would be affected. Somewhat surprisingly, we found that when the factor regarding the influence of phospholipid content on the membrane permeability for nucleotides and nucleotide precursors ($F_{PP}$) is turned up – even in a dramatic way, i.e., from its default value of 20–200, 2000, and $2 \times 10^4$, there is nearly no influence on the spread of GR (Fig 3-$F_{PP}$, orange symbols). In the model, according to the rule revealed in the original experimental work [9], the membrane permeability is assumed to be negatively related to phospholipid content in the membrane – in proportion to $1/(1 + F_{PP} \times RPM)$ (see Methods for details). Therefore, a possible cause for the little influence of $F_{PP}$ on GR's spread is concerning the "limited membrane-takeover" (i.e., with a low RPM). Indeed, if $P_{PF}$ is turned up to achieve a higher RPM (as mentioned above, refer to Fig B in S1 Text), we can see some effects (Fig C in S1 Text, the purple line) – but still rather limited. Finally, we turned to the weak version of $F_{PP}$, i.e., $F_{PPW}$. This factor is regarding the influence of phospholipid contents on the membrane permeability for precursors of nucleotide precursors and precursors of glycerophosphates – we increased it from its default value of 3 to a value of 3000 (in reality, this factor could not be very large because for such precursors, which should be quite small in molecular size, the permeability difference between the fatty acid membrane and phospholipid membrane could not be that large [2,42]), and see a more obvious decline in GR molecules (Fig 3-$F_{PP}$, the purple line).

But the GR protocells still only decline marginally (Fig 3-$F_{PP}$, purple circles), which means that there are fewer GR molecules in each GR protocell. Notably, for this case, RPM also declines for GR protocells (Fig D in S1 Text), which should result from the decrease of GR, as well as the less availability of its substrates – precursors of glycerophosphates. No matter how, the effect concerning $F_{PPW}$ means, an important reason why a quite high value of $F_{PP}$ alone would not obviously inhibit the spread of GR may be attributed to the remained availability of raw materials with a smaller size (thus with a greater permeability), as alternative resources.

### The subsequent spread of the ribozyme using more fundamental materials and the RNA species favoring the membrane transport

Indeed, when original raw materials for chemical synthesis in protocells are blocked by the membrane, more fundamental and permeable raw materials might have acted as alternative resources. However, as demonstrated above, the "initial membrane-takeover", which may have been caused by the emergence of one ribozyme (e.g., GR here) functioning in the pathway of phospholipid synthesis, should have been at a "low level". Then, the question becomes attractive whether such a "low level membrane-takeover" (thus with a relatively small influence on the membrane permeability) at the early stage, could, as supposed in the original experimental work [9], have driven the evolution concerning the arising of function for exploiting the more fundamental and permeable raw materials.

The answer is positive (see Fig 6a). In the simulation, at step $1 \times 10^4$, ten empty protocells are selected, each of which is inoculated with one NR molecule, one GR molecule, and one control molecule. Then the NR-GR protocells spread (the control does not spread). The solid circles and solid lines represent the case in which no influence of phospholipid content on the membrane's permeability is considered (i.e., $F_{PP}$ and $F_{PPW}$ are set to 0) throughout the whole simulation process, while the empty circles and dotted lines represent the case in which the "negative" influence of phospholipid content is turned on at step $3 \times 10^5$ (thereafter we can observe a little decrease of the NR-GR protocells and that of NR and GR molecules). For both cases, at step $6 \times 10^5$, ten NR-GR protocells are selected, each of which is inoculated with a molecule of nucleotide-precursor-synthetase ribozyme (NPR). The ribozyme is assumed to be able to catalyze the formation of nucleotide precursors from precursors of nucleotide precursors, which is more permeable. In the first case (i.e., without the negative influence of the phospholipid content), NPR cannot spread, whereas in the second case (i.e., with that negative influence), the NPR spreads (the black dotted line) – or say, the NR-GR-NPR protocells spread (the black empty circles). For snapshots of spatial distribution, see Fig Ea in S1 Text (notably, after the spread of NPR, its substrates, i.e., precursors of nucleotide precursors, are represented by the background yellow, are almost been exhausted – in the panel of step 2,000,000). In other words, the negative influence of phospholipid content on the membrane's permeability could indeed cause the thriving of the functional species exploiting more fundamental (thus more permeable) raw materials. To confirm that the spread of NPR is attributed to its function of exploiting precursors of nucleotide precursors, we turned off this function at step $1.4 \times 10^6$. As expected, we saw the vanishing of this species (Fig Fa in S1 Text).

Similar to the situation for exploring the spread of GR, to avoid the influence of unexpected random events of RNA degradation, here we selected ten NR-GR protocells and inoculated each with one molecule of NPR. The subsequent spread of NPR, in fact, already means that this RNA species may have emerged and become thriving in protocells. In reality, though it is impossible for NPR to have appeared simultaneously in so many protocells, it may have had chances to appear in protocells repeatedly especially considering the long time scale concerning the origin of life. For example, Fig Ga in S1 Text shows a modeling case that one NPR molecule is inoculated into one NR-GR protocell every $1 \times 10^5$ step, and NPR eventually spreads in the system.

Next, we turned to the plausibility of the emergence of a functional species that favors the membrane transport – as another strategy for "adapting to" the decreased membrane permeability, which was also proposed in the original experimental study [9]. An RNA species, named TR here, is assumed to be capable of favoring the membrane

PLOS Computational Biology

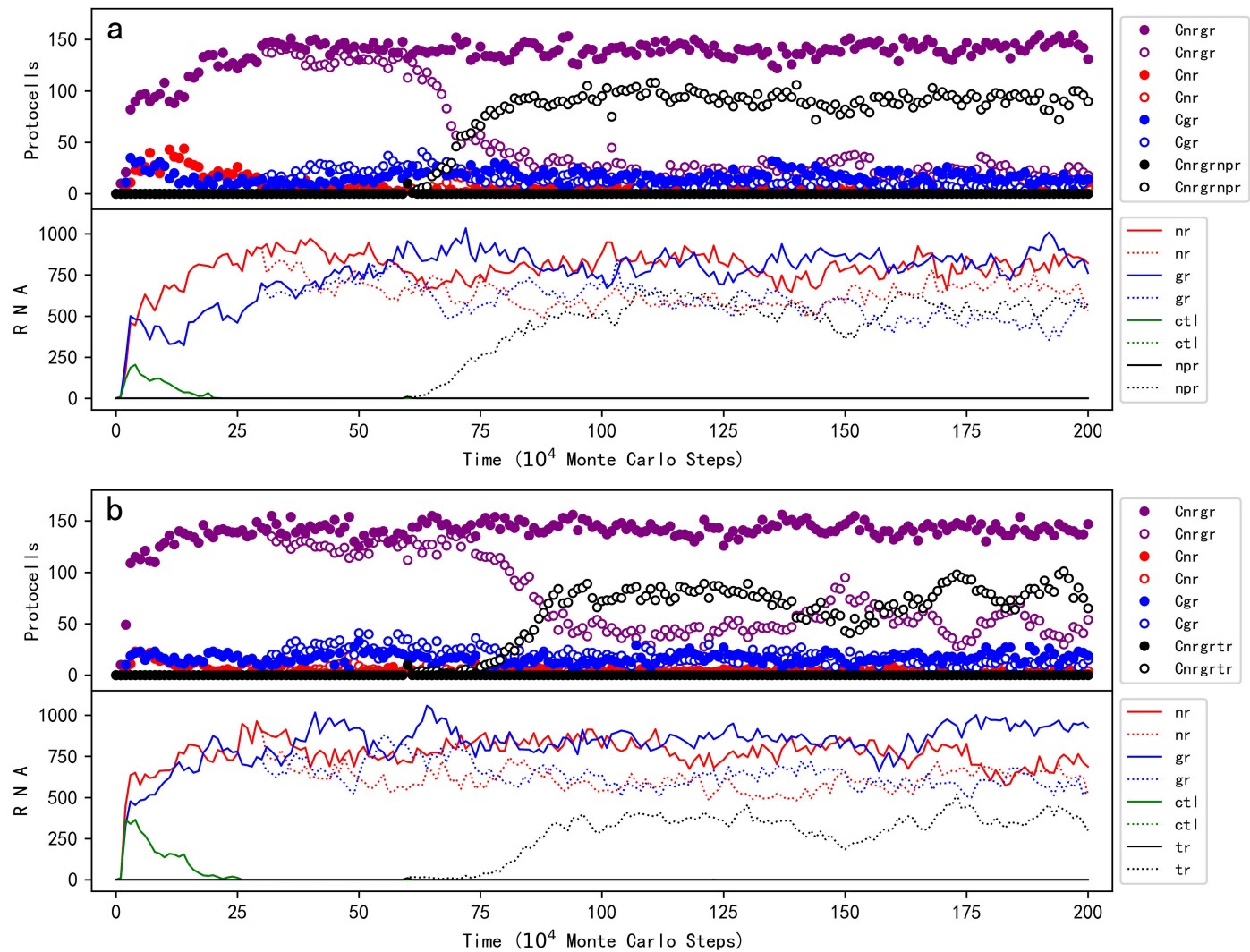

**Fig 6. The decreased permeability caused by phospholipid content in the membrane could drive the emergence of other functions in the protocells.** For each of the two subfigures, the upper panel shows the trend of relevant protocells while the lower panel demonstrates the trend of relevant RNA species. The legends identical to those included in Fig 2 have the same meaning, and additional ones are as follows: Cnrgrnpr — protocells containing NR, GR and NPR; Cnrgrtr — protocells containing NR, GR and TR; npr — NPR; tr — TR. At step $1 \times 10^3$, an empty fatty-acid protocell is inoculated. At step $1 \times 10^4$, ten empty protocells are selected (arbitrarily, the same below), each of which is inoculated with one NR molecule, one GR molecule, and one control molecule. (a) NPR, i.e., a ribozyme using more fundamental raw materials, would not spread if the influence of phospholipid content on membrane permeability is not assumed (solid circles and solid lines; $F_{PP}$ and $F_{PPW}$ are set to 0 throughout the simulation), but would spread when this influence is considered (empty circles and dotted lines; at step $3 \times 10^5$, $F_{PP}$ and $F_{PPW}$ are turned up to 30 and 3 respectively). In both cases, at step $6 \times 10^5$, ten NR-GR protocells are selected, each of which is inoculated with one NPR molecule. (b) TR, an RNA species favoring the membrane transport, would not spread if the influence of phospholipid content on membrane permeability is not assumed (solid circles and solid lines; $F_{PP}$ and $F_{PPW}$ are set to 0 throughout the simulation), but would spread when this influence is considered (empty circles and dotted lines; at step $3 \times 10^5$, $F_{PP}$ and $F_{PPW}$ are turned up to 30 and 3 respectively). In both cases, at step $6 \times 10^5$, ten NR-GR protocells are selected, each of which is inoculated with one TR molecule. $P_{NP} = 5 \times 10^{-6}$.

transport (see Discussion for a comment on this RNA species in a chemical context). As mentioned above, the permeable possibility of nucleotide precursors, i.e., the substrates of NR, has been assumed to be in proportion to $1/(1 + F_{PP} \times \text{RPM})$. Here, with the introduction of another factor, $F_{TR}$, the expression is changed to $1/[1 + F_{PP} \times \text{RPM}/$

$(1 + t \times F_{TR})$], where $t$ refers to the number of TR molecules in the protocell. That is, the more TR molecules there are in the protocell, the more permeable the membrane is. Notably, the increased permeability is here assumed to be in regard of both directions (inwards and outwards), in consideration that active transport should not yet have been achievable in such an early stage. The simulation showed that due to the negative influence of phospholipid content on the membrane's permeability, TR can spread in protocells (Fig 6b; for snapshots of spatial distribution, see Fig Eb in S1 Text), and when its function is turned off, it decreases and eventually vanishes (Fig Fb in S1 Text). Fig Gb in S1 Text shows a case in which one TR molecule is inoculated into one NR-GR protocell intermittently (every $1 \times 10^4$ step) and TR eventually becomes thriving in protocells.

## Discussion

In the present study, following the clues suggested by an experimental study from the Szostak group [9], we examined, through computer modeling, an evolution of protocells' membrane from the one composed of only single-chain amphiphiles like fatty acids towards the one containing double-chain amphiphiles like phospholipids, induced by simple physical effects. The former has been deemed to be the membrane of earliest protocells [1–3], whereas the latter is a membrane more approaching that of modern cells, which is more stable but less permeable. The simulation showed that such a membrane-takeover, though limited initially, could indeed occur on account of "stabilizing effects" caused by the increasing phospholipid content in the membrane [9], which is brought about by the emergence of a functional species favoring phospholipid synthesis in protocells (Fig 2). Subsequently, the reduced membrane permeability could trigger the emergence of an additional functional species which makes use of more fundamental (thus more permeable) raw materials, or a species facilitating the membrane transport (Fig 6) – both valid as supposed in the original experimental study [9].

In the modeling, as for the functional species favoring the synthesis of phospholipid, we adopted a ribozyme catalyzing the formation of glycerophosphates (GR), which appears to have been the bottle-neck reaction, and the resulting glycerophosphates is assumed to be able to reach the membrane and react with fatty acids there *in situ*, which seems to have been efficient even in a non-enzymatic way [4,5,25,26]. Another reason why we did not adopt a ribozyme catalyzing the latter reaction is that RNA is likely difficult to cope with reactions occurring on the membrane because of its polar skeleton – in reality, this function may have emerged after the advent of proteins. For example, in modern cells, the acylation reaction to form phospholipids (on the membrane) requires preexisting membrane-embedded enzymes [43,44]. Likewise, functions favoring the membrane transport seems also to have been implemented by proteins coming later. But there is also some evidence supporting RNA's potential role on membrane transport, e.g., see refs [45,46]. Anyway, at least to avoid a more complicated modeling involving proteins and amino acids, here we assumed an RNA species functioning this way (TR).

Our modeling study revealed some details regarding the membrane-takeover and relevant evolution. For instance, in the original experimental paper, as for the advantages of containing phospholipids in the membrane, it was pointed out that apart from phospholipids' effect of preventing the desorption of fatty acids, an additional mechanism is concerning "a decrease in the net efflux from the membrane due to the reduced fraction of the membrane surface area occupied by fatty acids" [9]. In other words, the fatty acids that have reacted to form phospholipids are "anchored" and would not participate in the desorption. Here we have figured out that this "fastening effect" is not absolute because phospholipids may also leave the membrane though much more difficult ($P_{PLM} << P_{FLM}$). By parameter analysis (Fig 3-$F_{PL}$ and -$P_{FLM}$; see also Fig A in S1 Text), we demonstrated how the two effects work together to support the thriving of the ribozyme favoring phospholipid-synthesis (GR).

As another detail, the initial membrane-takeover, most likely involving only one catalytic function in the pathway of phospholipid synthesis (e.g., GR here), should have been quite limited – i.e., with merely a low level of phospholipid content (Fig 5). A more thorough takeover is supposed to have occurred with the advent of other functions within the pathway (Fig B in S1 Text). That is, the membrane-takeover may have been a stepwise process in evolution. Notably, even before

the emergence of enzymatic function favoring the synthesis of phospholipids, there would still have been a level of phospholipid content in the membrane, though marginal (one may refer to the stage before the rising of GR, either in Fig 5a or 5b). This can be attributed to the non-enzymatic synthesis of phospholipids (quite inefficient, related to $P_{GF}$ and $P_{PF}$ in the model). Interestingly, it is shown that the GR, only via inducing such a limited membrane-takeover, can enjoy the benefit of phospholipids (i.e., stabilizing the membrane) and thrive in the system. Surprisingly as well, such a low level of phospholipid content, via its limited negative influence on the membrane permeability, is sufficient to trigger the emergence of the function for exploiting more fundamental and permeable raw materials (NPR) and that of the function for membrane transport (TR).

It should be noted here that there was an early theoretical paper [47] dealing with a topic related to the idea proposed in Budin and Szostak's experimental paper [9]. In that paper, Szathmáry suggested a scenario in which the membrane would have evolved towards becoming more impermeable to retain in protocells the building blocks synthesized by enzymatic reactions, and as a consequence, functions that favor across-membrane transportation may have then emerged. That is an evolutionary cascade from metabolism to membrane to transport mechanisms, while the experimental paper proposed an evolution order from membrane to metabolism and transport mechanisms. Certainly, the former scenario is another interesting one deserving modeling study in future.

Another related paper concerns a more recent modeling study [48], in which the evolution driven by the coupling of membrane and "proto-metabolism" was investigated. However, by "proto-metabolism" they mean that no enzymatic reaction is involved. That is, the genetically-encoded mechanism is not included in the scenario, and the relevant evolution is a "pre-Darwinian evolution". Compared with our model, the model used in this study is much more abstract, but notably, its method for dealing with the growth and division of protocells is more concrete than ours (i.e., considering the actual balance between vesicular surface area and internal volume). Additionally, their model is a "semi-empirical" one – i.e., with some assumptions just based on their own experimental data. This represents an effort to combine experimental work and theoretical work directly (see below for a discussion of the combination of experimental and theoretical studies in this field with a broader sense).

Remarkably, the series of evolutionary events associated with the membrane-takeover at an early stage of life, as demonstrated by the present modeling, exemplifies the scenario concerning the onset of Darwinian evolution, in which simple physical or chemical effects (e.g., here the decreasing efflux of fatty acids due to increasing phospholipid content in the membrane and the subsequent reduction of membrane permeability) may have driven the emergence of relevant functions. Additionally, as we have seen, albeit the degree of the effects might have been quite limited, new "inventions" could have still been induced – the power of Darwinian evolution is here clearly "witnessed".

In fact, such early evolutionary events of life belong to the field of biogenesis. This field, or named "the origin of life", is to a degree a problem of chemistry, which mainly addresses the environments and chemical mechanisms involved in the process [49–51] (generally referred to as prebiotic chemistry). On the other hand, however, it is undoubtedly also a problem of evolution, which involves the rules of the so-called "chemical evolution" and the subsequent early Darwinian evolution [52–54]. While experimental exploration has covered nearly the entire aspect regarding chemistry, it seems to be seriously constrained in the aspect of evolution. For instance, here, the clues for an early membrane-takeover came from an elegant experimental work of Budin and Szostak [9], which detected relevant simple physical mechanisms which might have led to the corresponding evolutionary events. However, it is at least up-to-now difficult for experimental researchers to follow up on those events (lab work's limitation in this respect could usually be attributed to the potential long time scale required in the evolution, as well as the complicated nature of these evolutionary events [16]). In contrast, theoretical modeling and associated computer simulation is apt at such exploration, i.e., on the plausibility of the suggested evolution and those underlying mechanisms [12–16,55]. It is expected that the combination of experimental and theoretical efforts like the one demonstrated in the present study would significantly enhance our understanding on those complex processes involved in the origin of life.

## Methods

### The events occurring in the model system

In each time step (i.e., Monte Carlo step), certain events may occur to molecules and protocells with defined probabilities (see Fig 7 for a schematic; refer to Table 1 for descriptions of the probabilities and other parameters). Only molecules within the same grid room can interact with each other. A molecule may move to an adjacent room (the related probability: $P_{MV}$) if there is no membrane in that direction, and may join or permeate through the membrane if there is one (amphiphiles and glycerophosphates may join while other molecules may permeate through, see below for details regarding relevant probabilities). A protocell may also move to an adjacent room ($P_{MC}$) (while pushing away molecules in that room).

Nucleotide-precursor's precursors may form nucleotide precursors in a non-enzymatic way ($P_{NPF}$) or catalyzed by NPR ($P_{NPFR}$). Nucleotide precursors may form nucleotides (randomly as A, G, C, or U) in a non-enzymatic way ($P_{NF}$) or catalyzed by NR ($P_{NFR}$), Glycerophosphate precursors may form glycerophosphates in a non-enzymatic way ($P_{GF}$) or catalyzed by GR ($P_{GFR}$). Nucleotide precursors, nucleotides, and glycerophosphates may also decay into their precursors ($P_{NPD}$, $P_{ND}$, and $P_{GD}$ respectively).

Nucleotides may join to form RNA via random ligation ($P_{RL}$). An RNA molecule may attract substrates (nucleotides or oligomers) ($P_{AT}$) via base-pairing with some error rate ($P_{FP}$), and substrates aligned on the template may be ligated ($P_{TL}$) – that is, the template-directed synthesis. The substrates or the full complementary chain may separate from the template ($P_{SP}$). Phosphodiester bonds within an RNA chain may break ($P_{BB}$) and thus the RNA molecule turns into two fragments. A nucleotide residue at the end of an RNA chain may decay into a nucleotide precursor ($P_{NDE}$).

In a grid room, if the quantity of amphiphiles (determined by the number of tails: a fatty acid is regarded as one, while a phospholipid is regarded as two) is larger than $L_{AM}$, they may aggregate at the boundary of the room and form a membrane ($P_{MF}$), thus creating a protocell. During the assembly process, all of the amphiphiles are incorporated into the membrane, while other molecules are located inside the protocell. Fatty acids may join or leave the membrane ($P_{FJM}$ and $P_{FLM}$ respectively), and phospholipids may also join or leave the membrane ($P_{PJM}$ and $P_{PLM}$ respectively). Nucleotide-precursor's precursors, nucleotide precursors, nucleotides and glycerophosphate precursors may permeate through the membrane ($P_{NPPP}$, $P_{NPP}$, $P_{NP}$ and $P_{GPP}$ respectively). Glycerophosphates may enter a membrane and react with fatty acids thereon *in situ* to form phosphatidic acids, i.e., phospholipids ($P_{PF}$). Phospholipids may decay into fatty acids and glycerophosphates, either within the membrane or out of the membrane ($P_{PDM}$ and $P_{PD}$ respectively). A protocell may fuse with another protocell in an adjacent grid room ($P_{CF}$), divide (with an offspring protocell occupying an adjacent grid room while pushing away molecules in that room) ($P_{CD}$), or break ($P_{CB}$) – resulting in the disassembly of its membrane components.

Notably, similar to our previous modeling work concerning the evolution of the RNA world, the energy problem is here not considered explicitly. For example, nucleotides and oligonucleotides are implicitly assumed to be activated – in particular, when they form from the degradation of RNA, they are assumed to be activated again immediately to be able to be reused in the further synthesis of RNA. Interestingly, such *in situ* activation within protocells has recently been shown possible by lab work [56,57]. In history, the energy source may have involved chemical energy in the hatchery of the primordial life, such as hydrothermal vents at the sea bottom [58–60] or hydrothermal fields on land [61,62], as supposed. Since the substrates are here assumed to be always "activated", the protocells in the model system are competing for materials but not energy – as mentioned already, the total materials in the system, including those related to RNA and the membrane of protocells are assumed to be limited (that is, $T_{NPPB}$, $T_{FB}$, and $T_{GPB}$, plus those material introduced by the rare events of inoculation). Certainly, in reality, competitions for materials and energy are both possible in Darwinian evolution.

Another issue, which is associated with the energy problem, concerns the synthetic reactions and the corresponding decay reactions of building blocks. In the model, the precursors (Npp, Np and Gp as depicted in Fig 7) are actually activated ones in the synthetic reactions (as a reactant) but inactivated ones in the decay reactions (as a product). In other words, a decay reaction here is actually not the reverse reaction of the corresponding synthetic reaction – both of them are actually irreversible. Notably, as a consequence, the ribozyme-catalyzed synthetic reactions (related to $P_{NPFR}$, $P_{NFR}$ and $P_{GFR}$) in the model are irreversible.

**a**

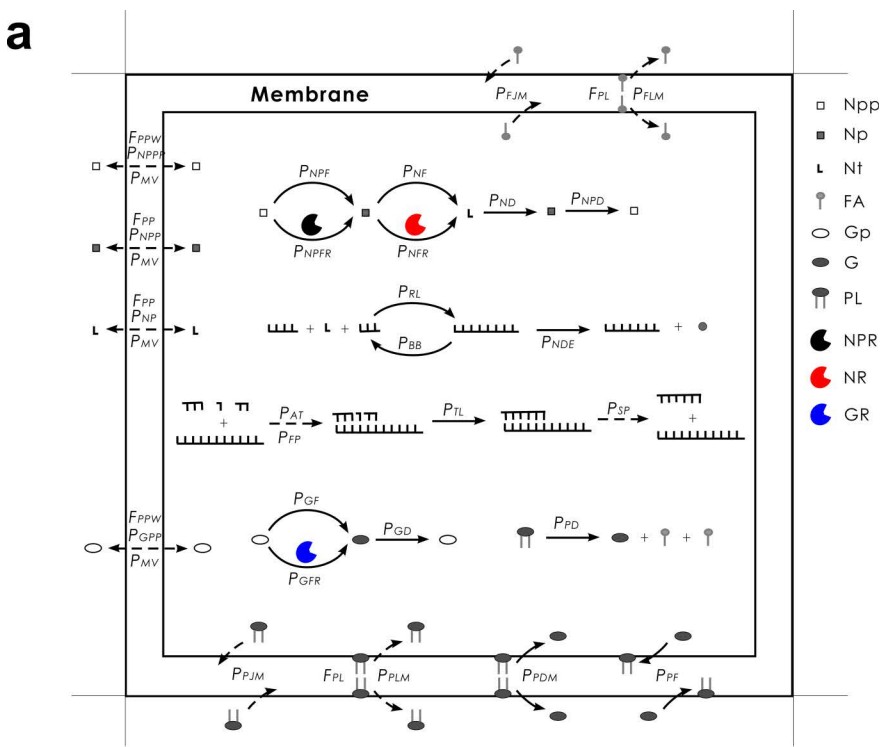

**b**

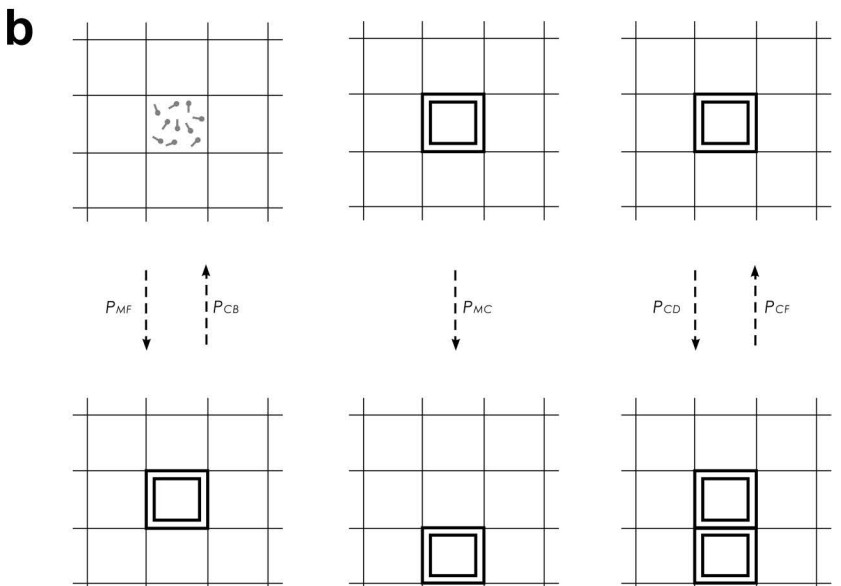

**Fig 7. Events occurring in the model system and associated parameters.** Solid arrows denote chemical reactions and dashed arrows represent other events. Legends: Npp—nucleotide-precursor's precursor; Np—nucleotide precursor; Nt—nucleotide; FA—fatty acid; Gp—glycerophosphate precursor; G—glycerophosphate; PL—phospholipid, i.e., phosphatidic acid here; NPR—nucleotide-precursor-synthetase ribozyme; NR—nucleotide-synthetase ribozyme; GR—glycerophosphate-synthetase ribozyme. The events occurring within a protocell are shown in (a), and the events concerning the behaviors of the protocells are depicted in (b), which adopts a smaller scale. For a naked room, there would be no membrane and associated events. Note that TR, i.e., the functional RNA species involved in the membrane transport, which functions in an abstract way in the model, is not depicted here; and there are a few parameters unsuitable or difficult to represent here (see text for detailed explanations).

## The setting of parameters

The parameters should be set according to some rules. For example, reactions catalyzed by ribozymes should be much more efficient than corresponding non-enzymatic reactions, so $P_{NFR} \gg P_{NF}$, $P_{GFR} \gg P_{GF}$, and $P_{NPFR} \gg P_{NPF}$. Template-directed ligation should be much more efficient than "random ligation", so $P_{TL} \gg P_{RL}$. The nucleotide residues within the chain are assumed to be protected from decay, whereas those at the end of the chain are only partially protected – i.e., may decay but at a rate lower than that of free nucleotides, i.e., $P_{NDE} < P_{ND}$. Considering experimental evidence on permeability, $P_{NPPP} > P_{NPP} \gg P_{NP}$ [2,42]. Because of the self-assembly feature of the membrane, $P_{MF} \gg P_{CB}$, $P_{FJM} \gg P_{FLM}$, and $P_{PJM} \gg P_{PLM}$. Phospholipids should more difficult to leave the membrane than fatty acids, so $P_{PLM} < P_{FLM}$. Phospholipids should be more difficult to decay within the membrane, so $P_{PDM} < P_{PD}$. The movement of molecules should be easier than protocells, so $P_{MV} > P_{MC}$. Nucleotides and RNA should be easier to degrade outside protocells (due to the higher water activity), so $F_{DO} > 1$; the influence of phospholipid content on the permeability of smaller molecules should be weaker than that of larger molecules, so $F_{ppw} < F_{pp}$ [2,42].

Obviously, the rules mentioned above are far from justifying the setting of that many parameters used here (Table 1). In fact, owing to our limited knowledge concerning prebiotic environments and chemistry, it is usually difficult to justify the parameter setting in the modeling studies concerning the origin of life. However, the evolution during the origin of life is remarkably characterized by the tendency from simplicity to complexity, which is a special, rare phenomenon in nature [52–54,63]. Therefore, any relevant hypothetic scene in the area (e.g., here, the speculation concerning the evolution of the protocell membrane), if supported by modeling, merits our attention. In this consideration, exploring parameter-setting in favor of the scene is valuable, which we called "parameter-exploration" in a way of "reverse modeling" (see ref [15] for a detailed discussion). In practice, here most parameters have been explored and adopted based on our experience in previous modeling studies concerning RNA-based protocells [22–24]. When manual testing was difficult, a machine learning-like approach was used to automatically explore the parameter space [15].

The default values listed in Table 1 were adopted to shape the cases for demonstrating our results. Actually, though the outcomes of the simulations may be influenced by the change of those "key parameters" (e.g., for GR's spreading, see Fig 3) and some of the other parameters (see Figs H and I in S1 Text, as explained in Box A in S1 Text), in general, they have turned out to be fairly robust against "moderate adjustments" of most of the parameters.

To avoid cumbersome computation, total materials ($T_{NPPB}$, $T_{FB}$, and $T_{GPB}$), "the lower limit number of amphiphiles to form a membrane" ($L_{AM}$), and the length of the characteristic domain for a functional RNA species ($L_{CDR}$) are set obviously smaller in scale than the corresponding situations in reality. Such simplifications are believed to be not in conflict with the fundamental mechanisms reflected in the modeling.

## Some detailed mechanisms concerning the implementation of the model

When the breaking site of an RNA chain is at a single-chain region, the breaking probability is $P_{BB}$. When the breaking site is within a double-chain region, the two parallel bonds may break simultaneously – but with a smaller probability, i.e., $P_{BB}^{3/2}$ (note that a probability has a value between 0 and 1). However, for the case of outside protocells, a factor, $F_{DO}$, is involved to consider the corresponding higher water activity ($F_{DO} > 1$): the breaking probability for a single-chain is $P_{BB} \times F_{DO}$, while that for a double-chain is $(P_{BB} \times F_{DO})^{3/2}$. The factor $F_{DO}$ also works in the situation of nucleotide decaying and nucleotide residue decaying at the end of RNA, i.e., $P_{ND} \times F_{DO}$ and $P_{NDE} \times F_{DO}$ respectively for the case of outside protocells. In fact, the introduction of this factor reflects our assumption about the difference between the interior and exterior of the protocells in the model system. Notably, a recent modeling study concerning RNA-based protocells suggested that the difference between the interior and exterior of the protocells may have been maintained owing to the intrinsic nature of the template-directed synthesis of RNA as a second-order autocatalytic process (i.e., both template and primer act as catalysts) [64].

The probability of the separation of the two strands of a duplex RNA is assumed to be $P_{SP}^{r}$, where $r = n^{1/2}$ and $n$ is the number of base pairs in the duplex. When $n = 1$, the probability would be $P_{SP}$. When $n$ increases, the separation of the

two strands would be more difficult. The introduction of 1/2 corresponds to the consideration that the self-folding of single chains may aid the dissociation of the duplex. It should be noted that in reality, the chain of a ribozyme may have been substantially longer than what we assumed in the model, and thus it is likely that the separation of the two strands was quite difficult. Although spontaneous self-folding of single chains might be of some help, RNA strand separation still seems to be an open problem in the field. Since the focus of the paper is not on this, here we do not intend to analyze it in detail.

The probability of membrane formation is assumed to be $1-(1-P_{MF})^x$, where $x = a-L_{AM}+1$ and $a$ is the number of amphiphiles (in quotient of tails, i.e., a fatty acid counts one whereas a phospholipid counts two, the same below) in the grid room. When $a$ is equal to $L_{AM}$ (the lower limit of the number of amphiphiles to form a protocell membrane), the probability of membrane formation is equal to $P_{MF}$. This assumption concerns the consideration that the more amphiphiles in a grid room, the more probable they would assemble to form a vesicle.

The probability of a fatty acid leaving the membrane is assumed to be $P_{FLM}/(y \times z)$, where $y = 1 + i/(b/2)^{3/2}$ and $z = 1 + F_{PL} \times RPM$. The item $y$ represents the consideration for the "osmotic pressure effect": a higher concentration of the inner impermeable ions would cause the protocell to be more swollen, and thus amphiphiles on the membrane are less likely to leave, as suggested by experimental work [65]. Wherein, $i$ is the quantity of inner impermeable ions, i.e. RNAs (measured by the number of nucleotide residues, the same below), and $b$ is the quantity of amphiphiles within the membrane. Then, $b/2$ (there are two layers in the membrane) is a "scale" representation of the surface area of the membrane and $(b/2)^{3/2}$ is a scale representation of the cellular space. Thus, $i/(b/2)^{3/2}$ is a representation of the concentration of the ions. The item $z$ represents the consideration for the phospholipid content's effect on preventing fatty acids from desorbing the membrane [9], wherein RPM refers to the ratio of phospholipids in the membrane (see the legend of Fig 5 for an explanation), and $F_{PL}$ is the factor representing the strength of this effect. Similarly, the probability of a phospholipid leaving the membrane is assumed to be $P_{PLM}/(y \times z)$, in which $y$ and $z$ are explained the same way. In other words, with the increase of phospholipid content, the membrane would be more stable, and any membrane components (including phospholipids themselves) would be less likely to leave the membrane [9,66].

It should be noted that, for the sake of simplicity in the model, we assume that a protocell occupies a grid room. This may represent a limitation regarding the incorporation of amphiphiles into the membrane, since any protocell, regardless of size, is only surrounded by the four sides of its occupied grid room. In other words, it does not take into account the fact that "the larger the membrane surface area, the greater the chance of the membrane encountering surrounding amphiphiles that might join it" – in reality, a growing membrane may have a chance to grow faster. Fortunately, this simplification constitutes a conservative assumption for our topic here – that is, if a growing membrane has a chance to grow faster, the ribozyme favoring phospholipid synthesis would enjoy a greater selective advantage to spread by favoring the growth of protocells that contain it.

The probability of a nucleotide permeating into a protocell is assumed to be $P_{NP} \times s/(u \times v)$, where $s = b/L_{AM}$, $u = 1 + F_{DE} \times i/(b/2)^{3/2}$, and $v = 1 + F_{PP} \times RPM$ (wherein, $i$, $b$ and RPM are explained in the same way as above). The item $s$ represents the consideration of the constraining effect of the cellular space on the influx of nucleotides. That is, when $b$ increases, meaning that the cellular space increases correspondingly, the probability of a nucleotide permeating into the protocell would become greater. The introduction of the item $u$ represents the consideration of the effect of Donnan's equilibrium [67], wherein $F_{DE}$ means the degree of this effect; simply put, RNAs, which are charged and impermeable, may suppress the incoming of permeable materials with the same charge, i.e., nucleotides here (see ref [22] for a detailed explanation). The introduction of the item $v$ represents the consideration of suppressing effect of phospholipid content on the permeability of the membrane, wherein $F_{PP}$ means the degree of this effect; that is, the permeation of nucleotides would decrease with the increase of phospholipid content [9]. Likewise, the probability of a nucleotide precursor permeating into a protocell is assumed to be $P_{NPP} \times s/(u \times v)$. With a little difference, the probability of a nucleotide-precursor's precursor permeating into a protocell is assumed to be $P_{NPPP} \times s/(u \times v')$, where $v' = 1 + F_{PPW} \times RPM$ and $F_{ppw}$ is a weak version of $F_{PP}$ – in consideration that the permeation of such small molecules should be suppressed with a less

extent [2,42]. The glycerophosphate precursor, i.e., glycerol here, is uncharged (thus no Donnan's equilibrium effect is considered) and small in molecular size (thus $F_{ppw}$ is adopted), so the corresponding permeating probability is $P_{GPP} \times s/v'$. Additionally, for the version of model involving the function of TR, the probability of a nucleotide precursor permeating into a protocell is assumed to be $P_{NPP} \times s/(u \times v'')$, where $v'' = 1 + F_{PP} \times RPM/(1 + t \times F_{TR})$ and $t$ is the number of TR molecules in the protocell. That is, the increase of TR molecules would enhance the membrane transportation of nucleotide precursors. Note that for the situations of permeating out from a protocell, the item of $s$ (concerning the cellular space) and $u$ (concerning Donnan's equilibrium) is not considered, e.g., for a nucleotide in a protocell, the probability of permeating outwards is simply $P_{NP}/v$.

The probability of protocell division is assumed to be $P_{CD} \times (1 - 2 \times L_{AM}/b)$, where $b$ is the quantity of amphiphiles within the membrane. When $b$ is no more than twice that of $L_{AM}$, the probability is no larger than 0, i.e., the protocell cannot divide. This assumption considers the fact that the larger the protocell, the more probable it would divide, on account of the physical instability.

The probability of the movement of an RNA molecule is assumed to be $P_{MV}/m^{1/2}$, where $m$ is the mass of the RNA, relative to a nucleotide. This assumption represents the consideration of the effect of the molecular size on the molecular movement. The square root was adopted here according to the Zimm model, concerning the diffusion coefficient of the polymer molecules in the solution [68].

(Note: Source codes of the simulation program can be obtained from GitHub—see Code availability statement. Besides the role of evidencing the reproducibility of the present study, the source codes present more details about the implementation of the model and may help readers to understand the simulation better).

## Supporting information

**S1 Text. Includes Figs A to I, Table A and Box A.**
(PDF)

## Author contributions

**Conceptualization:** Wentao Ma.

**Data curation:** Wentao Ma.

**Formal analysis:** Wentao Ma.

**Funding acquisition:** Wentao Ma.

**Investigation:** Wentao Ma.

**Methodology:** Wentao Ma, Chunwu Yu.

**Resources:** Chunwu Yu.

**Validation:** Wentao Ma.

**Writing – original draft:** Wentao Ma.

**Writing – review & editing:** Wentao Ma, Chunwu Yu.

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
