## [Decision Letter · Decision Letter 0]

PCOMPBIOL-D-25-00001

From experimental clues to theoretical modeling: Evolution associated with the membrane-takeover at an early stage of life

PLOS Computational Biology

Dear Dr. Ma,

Thank you for submitting your manuscript to PLOS Computational Biology. After careful consideration, we feel that it has merit but does not fully meet PLOS Computational Biology's publication criteria as it currently stands. Therefore, we invite you to submit a revised version of the manuscript that addresses the points raised during the review process.

Please submit your revised manuscript within 60 days Apr 25 2025 11:59PM. If you will need more time than this to complete your revisions, please reply to this message or contact the journal office at ploscompbiol@plos.org. Please include the following items when submitting your revised manuscript:

We look forward to receiving your revised manuscript.

Kind regards,

Joanna Slusky, Ph.D.

Academic Editor

PLOS Computational Biology

Tobias Bollenbach

Section Editor

PLOS Computational Biology

**Journal Requirements:**

At this stage, the following Authors/Authors require contributions: Wentao Ma, and Chunwu Yu. Please ensure that the full contributions of each author are acknowledged in the "Add/Edit/Remove Authors" section of our submission form.

5) We notice that your supplementary Figures, Tables, and information are included in the manuscript file. Please remove them and upload them with the file type 'Supporting Information'. Please ensure that each Supporting Information file has a legend listed in the manuscript after the references list.

1) State what role the funders took in the study. If the funders had no role in your study, please state: "The funders had no role in study design, data collection and analysis, decision to publish, or preparation of the manuscript.".

**Reviewers' comments:**

Reviewer's Responses to Questions

Reviewer #1: This paper has strong merits and some weakness. The strength is that it attempts to get as close to epperimental data as possible, without losing site of the evolutionary dimension. For the majority of the parameter values the Authors perform a careful choice. Nice work!

TECHNICAL ISSUES:

1. There are reactions that can be catalyzed by specific enyzmes. The uncatalzyed reactions are reversible. If so, the catalyzed reaction should also be reversible and faster in BOTH directions, hence the equilibrium constant for the same reaction should be the same. I think this will not lead to a qualitative change but will certainly make a quantitative difference. Since membrane growth and replication are sinks for the bulding blocks, the system will work, but less efficiently. The "protocells" are open systems, so the equilibria can shift with the degree of flow through the system, but opennness does not turn reversible reactions into irreversible ones. This is a consistency problem. I suggest the Authors do a control for ONE of the reactions to see the magnitude of the quantitative change. If there are surprises they have to redo everything, I am afraid.

2. A big simplification is how they treat RNA strand separation. It is still an open problem, unless the molecules are very short (as for von Kiedrowski origonucleotide replication). For most riboyzmes strand separation just will NOT work in the way they are modelling the process. 3D spontaneous folding might help somewhat, but it remains true that the totally annealed, complementary strands forming a duplex IS the STABLEST form (energy minimum). Having said this I think they can leave this as it is, since the focus of the paper is something else. But a remark on this is in order in the Discussion.

3. They define short necessary motifs for the three enzymatic functions. The small size is for cooputational reasons, but then it is very easy to go from one enzymatic function to another. Longer and more dissimilar motifs would require long and big simulations, for sure. But this brings me to another, interesting issue: the role of protocell fusion. The latter could turn out to be critical with long motifs, since then fusion could be necessary for bringing peices of the genome together (a bit ike the Fisher-Muller theory of sexual recombination). My concrete suggestion is to repeat at leat some of the runs by setting the fusion rate to zero to see what happens. Discuss the results.

CONCEPTUAL:

The title says "from experimental clues to theoretical modelling". Well said, but in historical reality it ought to something like "from theory to experiment to theory". They are missing two crucial theoretical references.

The first is Szathmary (2007). Coevolution of metabolic networks and membranes: the sceanrio oif progressive sequestration. This paper predates that of Szostak (Ref. 9. in the paper). That paper discusses the appearance of riboyzmes, progressive deccrease of permeability, the selection for "enyzamtization" of previously uncatalyzed reactios, and selecton for pemease function (referring to an exerimental RNA case). At any rate this paper is too close to be neglected. In reality one could say that the present manuscript has one theoretical root and one experimental root, but the former has been neglected. This should be remedied in the Introduction already.

The second paper that includes also quantitative modelling is Piedrafita et al. (2017) Permeability-driven selection in a semi-empirical protocell model: the roots of prebiotic systems evolution. Sci. Rep. 7, 3141. This is also a very serious neglect. That paper offers useful considerations as well as a dynamical model (although RNA replicators are not modelled). Compare it carefully with your approach in the Dsicussiion, and acknowledge key items that predate the present manuscript. Pay attention to the division problem that is just "written into" your present code.

After a major revision this manuscriot could become a strong paper.

Reviewer #2: In this manuscript, Monte Carlo simulations are set to simulate competing growth of protocells based on phospholipid metabolism, which was proposed based off of experimental studies by Budin and Szostak. In this model, catalyzed phospholipid synthesis would have driven the growth of protocell populations at the expense of others that produced fewer and no phospholipids. The mechanisms identified in that study were that phospholipids slowed the desorption rate of fatty acids from a membrane and that they themselves are insoluble, so they also proportionally increase adsorption vs. desorption surface area. This manuscript identifies both of these effects as potentially contributing to competitive protocell growth. They then explore other aspects of the emergence of phospholipid metabolism – increased selectivity for other relevant enzymes to maintain phospholipid catalysis and transporters to counter the reduced permeability rate of phospholipid membranes.

Overall, this is an interesting study, because this model for phospholipid-based competition has never been systematically investigated after its proposal. I would say the results here not surprising, but several of them are relevant, for example in identifying that both models of phospholipid effects on fatty acid dynamics could be relevant for competition. I do think the writing and presentation of the paper and its data could be substantially improved -making its data and figures more accessible to a broader audience while implementing a higher technical level of writing in the results and discussion.

1) The authors treat acylation reactions as trivial, with the limiting step to forming a PA being synthesis of glcerolphosphate. This is not the case; acylation is endergonic and requires a fatty acid to be activated first. Recent work they cited showing abiotic phospholipid synthesis all requires pre-activated fatty acid thioesters. Also, acylation of a naked glycerol to form a DAG also has the same growth phenotypes as PA, since the insolubility of the DAG also restricts efflux. Thus, I don’t understand the focus on a glycerolphosphate synthase ribozyme as the key step in this process.

2) Parameter for various functions (like fatty acid desorption rates) could be better constrained by experimental data, which exists. I understand the appeal of sweeping large ranges of parameter space to look for trends, but in this case we know some of these rates.

3) A good example of the above is the PFLM vs PPLM values assigned. A fatty acid is set to desorb 20 times slower than a phospholipid – in realty this more is more like a million fold or more based off of published data.

4) I find the figures 2,3,5, 6 to be extremely hard to parse for a broad audience

5) In the introduction, there is an incorrect statement/simplification of fatty acids and phospholipid systems (lines 47-52) . Fatty acids are more dynamic because they are soluble and contain small, less polar head groups. So exchange of fatty acids and flip flop across leaflets is much faster, and this can be utilized for growing membranes. The citations provided showing prebiotically plausible mechanisms of phospholipid synthesis and new interesting phenomenon in phospholipid GUVs does not change that.

6) The references study, which this manuscript is based on, has only two authors, so the correct citation in the is not Szostak and coworkers (plural). Budin and Szostak as the two authors or the Szostak group generally is more appropriate.

Reviewer #3: This paper considers various possible ribozymes that have a positive influence on the growth of protocells. The model shows that cells containing any of these ribozymes can be selected by evolution. The model is well motivated and the results are quite clear. I think the work is definitely publishable, although I have several suggestions for improvements and clarification.

The paper investigates the possible transition from membranes composed of fatty acids, to membranes composed of phospholipids. It is assumed that phospholipid synthesis is catalyzed by ribozymes. This is plausible, but maybe there could be other scenarios. In other words, could it be possible to have phospholipid membranes prior to the existence of RNA replication? Do we really know that growth and division of protocells must have been controlled by ribozymes? If we preferred a metabolism-first view of life, could we not also get a takeover of fatty acid membranes by phospholipids?

The rules used for formation and growth of membranes need to be explained more carefully. On Line 449, there is a minimum number of lipid molecules L_AM required on one lattice site to make a new cell. I assume that a cell instataneously assembles as a single event. What if there are more than L_AM molecules on the lattice site? Are the additional molecules included in the membrane or are they assumed to be inside the cell? What about any molecules of other kinds that are on the lattice site at the time the cell forms? Are these immediately assumed to be inside the cell?

It seems that if two cells fuse, this creates a single cell on one lattice site only. In other words, a single site can hold one cell of arbitrarily large size. Is this correct?

When fatty acids join or leave an existing membrane, do these come from neighboring lattice sites, or from the same site as the cell? If they come from the same site, are they assumed to be inside or outside the cell before they join the membrane? The squares representing the membranes are drawn around the whole lattice site. This suggests that all molecules on one site must be inside the cell, in which case new lipids must be added from neighboring cells. Line 433 says “see below for the situation of encountering a membrane” – but I cannot find where this is described.

For a fatty acid solution with concentration C, there should be some critical aggregation concentration C* above which membranes form spontaneously. For C < C*, there should be almost entirely dissolved molecules and no vesicles. For C > C* there will be a certain number of vesicles and a remaining number of dissolved molecules with a concentration equal to C*. Does the current model have this property? It would be useful to run a simulation with only fatty acids and no other molecules, and to measure the concentration of dissolved molecules and vesicles as a function of the total number of fatty acids, in order to show that a critical concentration exists.

If I understand correctly, membrane growth in this model requires diffusion of a dissolved lipid from one site onto a neighbouring site that already contains a cell. The rate of addition of molecules to the membrane therefore appears to be independent of the current membrane area. I am assuming that each lipid in a membrane has the same probability of leaving the membrane, so the rate of molecules leaving the membrane is proportional to the current area of the membrane. If there were no cell division or fusion in the model, cells would reach a steady state size where loss of lipids balances addition. In reality, cells have different membrane areas. We would expect the rate of addition of molecules to a cell of area A to be proportional to A. If both addition and loss of lipids were proportional to current area, then the area would either increase or decrease exponentially, according to whether addition or removal were larger. This seems to be an important difference.

The factor y on line 533 represents the osmotic pressure effect observed in ref 60, where it is observed that cells containing RNAs with no function gain lipids at the expense of empty vesicles. This experimental effect appears important, and it would be useful to demonstrate it in the model – i.e. to show that if you start with a mixture of cells containing random non-functional RNA templates and cells containing no RNA, the cells containing RNAs will multiply and the empty cells will die.

The functional forms for the factors y and z on line 534 seem very qualitative. I agree that you want the leaving rate of the fatty acids to decrease when the fraction of phospholipids goes up and you want the leaving rate of the lipids to decrease when there is an osmotic pressure (or the addition rate to increase). But there is no “physics” in these functions. I would have preferred it if these functions could be derived from consideriations of the energetic stability of the membrane, rather than just making up arbitrary functions. I also note that the scaling b^3/2 on line 540 is only true if the cell is spherical. In reality cells cannot be spherical. Spherical cells cannot divide because there would be insufficient lipid to enclose two smaller spheres of half the volume. There is no physics of cell division in this model – it is supposed that there is a constant probability of division of all cells, but in reality there must be some dependence on volume, area and shape of the cells. I see that the main point of this paper is to demonstrate selection for ribozymes with useful function. I expect that this will occur whatever the details of the cell division process, but I am pointing out that a lot of interesting aspects of cell division behaviour are ignored in this model.

The model has rules for RNA polymerization and templating. I am presuming that these reactions happen both inside and outside cells. In other words, there is nothing that differentiates the inside and outside of the cell. In reality we would like a cell to have an active autocatalytic reaction inside, while the environment remains inactive. We do not want autocatalysis to start spontaneously in the environment otherwise there would be no difference between inside and outside. See recent paper of Sanders et al https://journals.aps.org/pre/abstract/10.1103/PhysRevE.111.014424 . In that paper we have shown that if templating is fast, but spontaneous synthesis of oligomers from monomers is slow, then it is possible for a cell that begins with oligomers to maintain templating inside the cell while there is no initiation of templating outside the cell. It takes a rare event to create a cell that contains sufficient oligomers to initiate the templating process, but once initiated, RNA synthesis continues. Thus RNA templating can play the role of a metabolism that maintains the difference between the inside and outside of a cell. For the model used in the current paper I am therefore interested in whether templating can initiate if there are only monomers to start with. It would seem reasonable to set polymerization and templating rates such that templating is maintained inside cells that begin with oligomers but is not initiated outside cells where there are no oligomers.

**Have the authors made all data and (if applicable) computational code underlying the findings in their manuscript fully available?**

Reviewer #1: Yes

Reviewer #2: **No: **

Reviewer #3: None

PLOS authors have the option to publish the peer review history of their article (what does this mean? ). If published, this will include your full peer review and any attached files.

**Do you want your identity to be public for this peer review?** For information about this choice, including consent withdrawal, please see our Privacy Policy .

Reviewer #1: No

Reviewer #2: No

Reviewer #3: No

**Figure resubmission:**

**Reproducibility:**



---

## [Decision Letter · Decision Letter 1]

PCOMPBIOL-D-25-00001R1

From experimental clues to theoretical modeling: Evolution associated with the membrane-takeover at an early stage of life

PLOS Computational Biology

Dear Dr. Ma,

Thank you for submitting your manuscript to PLOS Computational Biology. After careful consideration, we feel that it has merit but does not fully meet PLOS Computational Biology's publication criteria as it currently stands. Therefore, we invite you to submit a revised version of the manuscript that addresses the points raised during the review process.

Please submit your revised manuscript within 30 days Jul 22 2025 11:59PM. If you will need more time than this to complete your revisions, please reply to this message or contact the journal office at ploscompbiol@plos.org. Please include the following items when submitting your revised manuscript:

We look forward to receiving your revised manuscript.

Kind regards,

Joanna Slusky, Ph.D.

Academic Editor

PLOS Computational Biology

Tobias Bollenbach

Section Editor

PLOS Computational Biology

**Reviewers' comments:**

Reviewer's Responses to Questions

Reviewer #1: The paper has been duly revised. I think that there is a problem in the response to Reviewer 3. If growth of the suface is not proportional to the membrane surface are but loss from the mebrane is, then indeed one should get a steady state.

Consider the equation

dA/dt = k - b A

where A is the surface. It is trivial that the equilibrium will be A = k/b.

If growth is also proportional to surface then we have

dA/dt = k A - b A = A(k-b),

and indeed if k>b then A growth exponentially.

Reviewer #3: The replies to the previous questions seem good. I have no further issues.

**Have the authors made all data and (if applicable) computational code underlying the findings in their manuscript fully available?**

Reviewer #1: Yes

Reviewer #3: Yes

PLOS authors have the option to publish the peer review history of their article (what does this mean? ). If published, this will include your full peer review and any attached files.

**Do you want your identity to be public for this peer review?** For information about this choice, including consent withdrawal, please see our Privacy Policy .

Reviewer #1: No

Reviewer #3: No

**Figure resubmission:**
---

## [Editor Report · Decision Letter 2]

Dear Dr. Ma,

We are pleased to inform you that your manuscript 'From experimental clues to theoretical modeling: Evolution associated with the membrane-takeover at an early stage of life' has been provisionally accepted for publication in PLOS Computational Biology.

Best regards,

Joanna Slusky, Ph.D.

Academic Editor

PLOS Computational Biology

Tobias Bollenbach

Section Editor

PLOS Computational Biology

---

## [Editor Report · Acceptance letter]

PCOMPBIOL-D-25-00001R2

From experimental clues to theoretical modeling: Evolution associated with the membrane-takeover at an early stage of life

Dear Dr Ma,

I am pleased to inform you that your manuscript has been formally accepted for publication in PLOS Computational Biology. Your manuscript is now with our production department and you will be notified of the publication date in due course.

With kind regards,

Anita Estes
